# The Association of Tumor Immune Microenvironment of the Primary Lesion with Time to Metastasis in Patients with Renal Cell Carcinoma: A Retrospective Analysis

**DOI:** 10.3390/cancers14215258

**Published:** 2022-10-26

**Authors:** Kazutoshi Fujita, Go Kimura, Toyonori Tsuzuki, Taigo Kato, Eri Banno, Akira Kazama, Ryo Yamashita, Yuto Matsushita, Daisuke Ishii, Tomoya Fukawa, Yuki Nakagawa, Tamaki Fukuyama, Fumikazu Sano, Yukihiro Kondo, Hirotsugu Uemura

**Affiliations:** 1Department of Urology, Kindai University Faculty of Medicine, Osaka 589-8511, Japan; 2Department of Urology, Nippon Medical School, Tokyo 113-8603, Japan; 3Department of Surgical Pathology, Aichi Medical University Hospital, Nagakute 480-1195, Japan; 4Department of Urology, Osaka University Graduate School of Medicine, Osaka 565-0871, Japan; 5Department of Urology, Division of Molecular Oncology, Niigata University Graduate School of Medical and Dental Sciences, Niigata 951-8510, Japan; 6Division of Urology, Shizuoka Cancer Center, Shizuoka 411-8777, Japan; 7Department of Urology, Hamamatsu University School of Medicine, Hamamatsu 431-3192, Japan; 8Department of Urology, Kitasato University of Medicine, Sagamihara 252-0374, Japan; 9Department of Urology, Tokushima University Graduate School of Biomedical Sciences, Tokushima 770-8503, Japan; 10Clinical Development Division, Chugai Pharmaceutical Co., Ltd., Tokyo 103-8324, Japan; 11Medical Affairs Division, Chugai Pharmaceutical Co., Ltd., Tokyo 103-8324, Japan

**Keywords:** TIME (tumor immune microenvironment), synchronous, metachronous, mRCC (metastatic renal cell carcinoma), PD-L1, immunophenotype

## Abstract

**Simple Summary:**

The association between the tumor immune microenvironment (TIME) of primary lesions and time to metastasis remains unknown. The aim of our retrospective study was to investigate the differences in the TIME of primary lesions based on time intervals to metastasis, mainly between the synchronous group (SG; metastasis within 3 months) and metachronous group (MG; metastasis after 3 months), and its association with clinicopathological parameters in patients with metastatic renal cell carcinoma (mRCC). SG showed more immunogenic feature of TIME (PD-L1 positivity, CD8+ TIL infiltration) and poor prognostic pathological features (WHO/ISUP grade 4, necrosis, lymphovascular invasion, infiltrative growth pattern, and sarcomatoid differentiation). In addition, we observed that the time to metastasis differed by TIME characteristics (PD-L1 status, immunophenotype), which were associated with the WHO/ISUP grade. The TIME of primary lesions could affect the time to metastasis.

**Abstract:**

Biological or immunological differences in primary lesions between synchronous and metachronous metastatic renal cell carcinoma (mRCC) have been reported. However, the association between the tumor immune microenvironment (TIME) of primary lesions and time to metastasis remains unknown. We investigated the differences in the TIME of primary lesions based on time intervals to metastasis, mainly between the synchronous group (SG; metastasis within 3 months) and metachronous group (MG; metastasis after 3 months), and its association with clinicopathological parameters in patients with mRCC. Overall, 568 patients treated first-line with vascular endothelial growth factor receptor inhibitors comprised the analysis population (SG: N = 307 [54.0%]; MG: N = 261 [46.0%]). SG had a higher proportion of patients with poor prognostic pathological feature tumors: WHO/ISUP grade 4, necrosis, lymphovascular invasion, infiltrative growth pattern, and sarcomatoid differentiation. Regarding the TIME, more immunogenic features were seen in SG than MG, with a higher PD-L1 positivity and a lower proportion of the desert phenotype. This is the first study to examine the differences in the TIME of primary lesions in patients with mRCC based on the time intervals to metastasis. The TIME of primary lesions could affect the time to metastasis.

## 1. Introduction

Approximately two thirds of patients with renal cell carcinoma (RCC) have localized disease and undergo radical nephrectomy with curative intent. However, 20% of these patients develop distant metastasis, that is, metachronous metastasis. The remaining one third of the patients with RCC present with metastasis at diagnosis, that is, synchronous metastasis. The prognosis of synchronous and metachronous metastatic RCC (mRCC) is different [1,2], with the time from diagnosis to treatment of less than 1 year being one of the most important risk factors for poor survival in both International Metastatic Renal-Cell Carcinoma Database Consortium (IMDC) and Memorial Sloan-Kettering Cancer Center (MSKCC) prognostic models for mRCC [3,4].

Previous reports have documented biological or immunological differences in primary lesions between synchronous and metachronous mRCC [5,6]. However, the association between the tumor immune microenvironment (TIME) of the primary lesion, the source of metastasis, and the time to metastasis has not been studied.

Furthermore, the TIME also influences the efficacy of immune-oncology (IO) drugs, such as anti–programmed cell death protein 1 (PD-1)/programmed death ligand 1 (PD-L1) and anti–cytotoxic T-lymphocyte–associated protein 4 (CTLA-4) antibodies [7,8]. Currently, IO combination therapies, including IO plus IO (e.g., nivolumab plus ipilimumab) or IO plus vascular endothelial growth factor receptor inhibitor (VEGFRi; e.g., pembrolizumab plus axitinib, avelumab plus axitinib, nivolumab plus cabozantinib, and pembrolizumab plus lenvatinib), are the standard first-line (1L) therapy for patients with mRCC [9]. 

Development of IO therapy is expanding into the area of adjuvant RCC [10]. Pembrolizumab showed significant improvement in disease-free survival (DFS) in the high-risk RCC which included M1NED, defined as resection of the primary tumor and solid, isolated, soft-tissue metastases [11]. Thus, understanding the relationship between time to metastasis and the TIME of the primary lesion will help in selecting the optimal therapy. 

The TIME can be described by the presence or absence of immune cells in the tumor area, the location of immune cells (infiltrated or excluded), the type of immune cells (anti-tumor immunity or immune suppressive cells, such as regulatory T cells and myeloid-derived suppressor cells), and the T cell status (activation, exhaustion, dysfunction) [12]. RCC is one of the immunogenic tumors, known as having a complex tumor microenvironment with immune suppressive cells [13]. We examined how PD-L1 expression and the immunophenotype of tumor-infiltrating immune cells in primary lesions differ by time to metastasis and its association with clinicopathological parameters in patients with mRCC. 

## 2. Materials and Methods

### 2.1. Study Design and Outcomes

This report is based on an additional analysis of a dataset from a previous multicenter, retrospective study that compared overall survival (OS) by PD-L1 expression status in patients with recurrent or mRCC who had received systemic therapy (the ARCHERY study [14]). We investigated the differences in the TIME of the primary lesions between synchronous and metachronous mRCCs with different time intervals to metastasis.

### 2.2. Patients

A total of 770 patients with recurrent or mRCC who had started systemic therapy between January 2010 and December 2015 at 29 institutions in Japan were enrolled in the ARCHERY study. Of these, 381 patients underwent radical nephrectomy and 389 underwent cytoreductive nephrectomy. Only patients whose formalin-fixed paraffin-embedded nephrectomy specimens could be obtained were registered in this retrospective study. Patients with other coexisting malignancies or those treated with checkpoint inhibitors as 1L systemic therapy were excluded [14].

An exploratory study was conducted using the ARCHERY database (the JEWEL study), in which patients with an unknown International Society of Urologic Pathologists (ISUP) grade in the ARCHERY study were excluded (N = 4). To improve the homogeneity of the population, we focused on patients from the ARCHERY study who had received 1L tyrosine kinase inhibitor (TKI; ie, VEGFRi) therapy (N = 569). The analysis population consisted of 568 patients, excluding one of the TKI-treated patients whose time to metastasis was not specified (Appendix A).

This study was registered in the UMIN Clinical Trials Registry (JEWEL study, UMIN000043415) and conducted with the approval of the institutional review board of the 29 study facilities. Furthermore, we obtained approval from the institutional review board of MINS (Reference number: 210204; approval date: 18 February 2021), a nonprofit organization. Informed consent was obtained from all participants, and the study was conducted in accordance with the Declaration of Helsinki.

### 2.3. Assessment of Histology and Immune Status

Hematoxylin-eosin–stained specimens were evaluated by two central pathologists. PD-L1 expression in tumor-infiltrating immune cells was analyzed using the VENTANA SP142 assay (Ventana Medical Systems, Inc., Tucson, AZ, USA, #740-4859). According to PD-L1 expression on immune cells (ICs), patients were classified as either PD-L1 negative (IC0 [PD-L1-expressing IC < 1%]) or PD-L1 positive (IC1 [IC: ≥1% but <5%], IC2 [ IC: ≥5% but <10%], or IC3 [IC: ≥10%]) [15]. Three distinct immunophenotypes—inflamed (many CD8-positive T cells diffusely infiltrate into tumor cell nests), excluded (CD8-positive T cells infiltrate around tumor cell nests but not into them), and desert (no or few CD8-positive T cells infiltrate around and/or into tumor cell nests)—were identified using CD8 immunostaining [16]. The immunohistochemical assessments were performed during the central pathological review, and PD-L1 expression was evaluated independently by two central pathologists.

### 2.4. Time to Metastasis

Time to metastasis was defined as the time from the date of the “initial diagnosis” to the “date of metastasis.” Information on whether the patient had distant metastasis at the time of initial diagnosis was collected from electronic data capture. For patients who did not have distant metastasis at the time of initial diagnosis, the “date of distant metastasis” was defined as the “date of recurrence.” For patients who had distant metastasis at the time of initial diagnosis, the “date of distant metastasis” was defined as the “date of initial diagnosis.” 

Patients who had metastasis at the time of nephrectomy and patients who experienced metastasis within 3 months after nephrectomy were categorized as the “synchronous group” (SG). Patients who experienced metastasis 3 months after nephrectomy were categorized as the “metachronous group” (MG). There is no consensus on when the onset of metastasis should be considered metachronous—previous reports have referred to onset after the time of diagnosis (>0 months) [17], after 3 months [2], and after 6 months [5]; the 3-month cutoff was chosen for this study. Patients in the MG were further categorized into four subgroups based on the time interval from initial diagnosis to recurrence: 3–12 months, >12–24 months, >24 months–5 years, and >5 years after nephrectomy.

### 2.5. Statistical Analysis

Baseline characteristics were summarized by time-to-metastasis categories, and standardized differences (SDs) were calculated as a measure of the difference between the two groups. A logistic model was constructed with categorized time to metastasis (SG/MG) as the response variable and baseline characteristics as the candidate explanatory variables to explore baseline variables relevant to time-to-metastasis categorization. Stepwise selection began with no variables selected and was used for explanatory variable selection in the multivariable logistic model. Variables that met the entry or removal criteria were added or removed until a stable set of explanatory variables was obtained. The Wald test was used to determine explanatory variables in the multivariable logistic model, with two significance levels: α = 0.05 for variable addition and α = 0.05 for removal. Time-to-metastasis distributions were estimated using the Kaplan-Meier method for several baseline characteristics, with confidence intervals (CIs) of the median estimated using the Brookmeyer–Crowley method. Univariate Cox proportional hazards models were used to estimate the hazard ratios (HRs) and CIs. The mean, difference, and CIs of time to metastasis were also calculated as all patients were not censored. All *p*-values provided were interpreted in a descriptive manner.

## 3. Results

### 3.1. Baseline Characteristics of Patients with Synchronous and Metachronous mRCC

In total, 568 patients with mRCC were included in the analysis. Of these, 307 (54.0%) had synchronous metastasis and 261 (46.0%) developed metastasis 3 months after nephrectomy. The baseline characteristics at the time of initial diagnosis of SG and MG are shown in Table 1. The median age was 64.0 [range: 23, 87] years in SG and 64.0 [30, 85] years in MG. The distribution of World Health Organization (WHO)/ISUP grades was different between SG (grade 1/2: 29.6%, grade 3: 38.1%, and grade 4: 32.2%) and MG (grade 1/2: 47.5%, grade 3: 37.5%, and grade 4: 14.9%). The proportion of patients with grade 1/2 tumors was lower in SG than in MG (29.6% vs. 47.5%, SD: −0.4), while the proportion of patients with grade 4 tumors was higher in SG than in MG (32.2% vs. 14.9%, SD: 0.4). The proportions of patients with other poor prognostic pathological features (necrosis [50.8% vs. 34.9%, SD: 0.3], lymphovascular invasion (LVI) [30.6% vs. 19.2%, SD: 0.3], sarcomatoid differentiation [16.3% vs. 5.4%, SD: 0.4], and infiltrative growth pattern [28.3% vs. 21.1%, SD: 0.2]) were higher in SG than in MG. 

SG showed higher PD-L1 positivity (50.2% vs. 30.7%, SD: 0.4) than MG. Regarding immunophenotypes, the proportion of the desert phenotype was lower (35.5% vs. 51.0%, SD: −0.3), while that of the inflamed phenotype was higher (9.4% vs. 3.4%, SD: 0.2) (Table 1) in SG compared with MG.

### 3.2. Multivariable Logistic Regression Analysis of Metachronous/Synchronous Metastasis

The multivariable logistic regression model selected PD-L1 expression, WHO/ISUP grade, and LVI as explanatory variables (Table 2). 

Although the distribution of immunophenotype was different between SG and MG (Table 1), the immunophenotype was not selected as an explanatory variable in the model, possibly because of the strong association between immunophenotype and PD-L1 expression and WHO/ISUP grade. PD-L1 positivity showed a distinct difference (desert vs. excluded vs. inflamed: 9.1% vs. 62.5% vs. 84.2%, respectively). The proportions of patients with WHO/ISUP grade 1/2 (52.5% vs. 29.5% vs. 7.9%) and grade 4 (14.5% vs. 27.4% vs. 63.2%) tumors were also considerably different (Appendix A).

### 3.3. TIME and Time to Metastasis

The median time to metastasis was 6.5 months (95% CI: 2.5, 8.3) in PD-L1–negative patients and 0 months (95% CI: not applicable [NA], NA) in PD-L1–positive patients (HR: 1.57, 95% CI: 1.33, 1.86, Figure 1A).

The median time to metastasis was 6.6 months (95% CI: 0.0, 9.0) for desert, 0 months (95% CI: 0.0, 0.8) for excluded, and 0 months (95% CI: NA, NA) for inflamed immunophenotypes (HR desert vs. excluded: 1.35 [95% CI: 1.13, 1.60], HR desert vs. inflamed: 1.77 [95% CI: 1.26, 2.50], Figure 1B).

PD-L1 positivity and immunophenotype of the five subgroups categorized according to the time interval from initial diagnosis to metastasis (≤3 months, >3–12 months, >12–24 months, >24 months–5 years, and >5 years) are shown in Figure 2 and Table 3.

PD-L1 positivity was highest for time to metastasis ≤ 3 months (50.2%) and tended to decrease with increasing time of metastasis (>3–12 months: 42.0% and >12–24 months: 23.1%). However, it was similar in the groups with confirmed metastasis after 12 months (>12–24 months: 23.1%, >24 months–5 years: 27.9%, and >5 years: 23.8%; Figure 2A and Table 3). With a 12-month cutoff point for the time to metastasis, PD-L1 positivity was 48.5% in the ≤12 months group (N = 388) and 25.6% in the >12 months group (N = 180). The SD was 0.5 (Appendix A).

The total number of patients with the inflamed phenotype was 38 (6.7%); therefore, the number of patients within each category was also quite small. The proportion of patients with the desert phenotype tended to increase with increasing time to metastasis (≤3 months: 35.5%, >3–12 months: 40.7%, >12–24 months: 50.0%, >24 months–5 years: 59.3%, and >5 years: 54.8%; Figure 2B and Table 3).

### 3.4. Pathological Features and Time from Initial Diagnosis to Metastasis

The pathological characteristics of the five subgroups categorized according to the time interval from initial diagnosis to metastasis (≤3 months, >3–12 months, >12–24 months, >24 months–5 years, and >5 years) are shown in Table 3.

The proportion of patients with different LVI and WHO ISUP grades, which were suggested to be related to metachronous/synchronous disease based on the multivariate analysis, showed a similar trend.

The proportion of patients with LVI differed between time to metastasis < 12 months and >12 months (≤3 months: 30.6% and >3–12 months: 28.4% vs. >12–24 months: 13.5%, >24 months–5 years: 15.1%, and >5 years: 16.7%). The proportion of patients with grade 1/2 tumors was also different when time to metastasis was <12 months vs. >12 months (≤3 months: 29.6% and >3–12 months: 25.9% vs. >12–24 months: 50.0%, >24 months–5 years: 59.3%, and >5 years: 61.9%; Table 3). 

With a 12-month cutoff point for the time to metastasis, the proportion of patients with LVI was 30.2% in the ≤12 months group (N = 388) and 15.0% in the >12 months group (N = 180). The SD was 0.4. The proportions of patients with grade 1/2 (28.9% vs. 57.2%, SD: −0.6) and grade 4 (29.9% vs. 12.2%, SD: 0.4) tumors were also distinctly different (Appendix A).

### 3.5. Clinical Characteristics at the Time of 1L Treatment

The clinical characteristics at the time of 1L treatment are shown in Table 4. The median age at the time of initial diagnosis was different between the groups: 64.0 [range: 23, 87] in SG and 68.0 [31, 89] in MG. The distribution of the IMDC risk groups was different between SG (favorable: 3.6%, intermediate: 64.8%, and poor: 31.6%) and MG (favorable: 39.5%, intermediate: 51.3%, and poor: 9.2%).

Comparing the sites of metastasis in SG and MG, a higher proportion of lung (SG vs. MG, 69.1% vs. 55.6%, SD: 0.3) and lymph node (26.4% vs. 17.6%, SD: 0.2) involvement was observed in SG, while a higher proportion of pancreas (2.6% vs. 6.5%, SD: −0.2) involvement was observed in MG (Appendix A).

The median OS after 1L VEGFRi treatment was 29.5 months (95% CI: 25.1, 32.5) in SG and 44.2 months (95% CI: 36.5, 51.1) in MG (HR, 0.74 [95% CI: 0.60, 0.91]) (Appendix A).

## 4. Discussion

Metastasis is seen in synchronous mRCC at the time of diagnosis, while metachronous metastasis develops after nephrectomy for localized RCC. In this study, we investigated the differences in PD-L1 expression and immunophenotype of tumor-infiltrating immune cells in primary lesions, mainly between SG and MG, based on time intervals to metastasis and its association with clinicopathological parameters in patients with mRCC.

### 4.1. The Difference in Baseline Characteristics between SG and MG

At the time of initial diagnosis, a higher proportion of poor prognostic pathological features were seen in SG compared with MG (WHO/ISUP grade 4, necrosis, LVI, infiltrative growth pattern, and sarcomatoid differentiation). Similarly, regarding the TIME, more immunogenic features were observed in SG than in MG: PD-L1 positivity (50.2% vs. 30.7%, SD: 0.4), proportion of desert phenotype (35.5% vs. 51.0%, SD: −0.3), and inflamed phenotype (9.4% vs. 3.4%, SD: 0.2). These pathological/immunological differences between SG and MG were consistent with previous reports and support poor prognosis in SG compared with MG [2,5,6].

### 4.2. Association between TIME and Time to Metastasis

Several previous reports have shown an association between TIME and pathological grade; T-cell infiltration with the expression of several checkpoint proteins (e.g., PD-1 and PD-L1/2) was observed to be associated with a more aggressive phenotype, higher pathological grade, and sarcomatoid differentiation [18,19,20,21]. In addition, T-cell infiltrating tumor types with checkpoint protein expression were considered to have a worse prognosis than those with the immune desert phenotype in both metastatic and localized RCC [20,22,23]. We observed that PD-L1 positivity was associated with a shorter time to metastasis. Among immunophenotypes, the desert phenotype was associated with a longer time to metastasis than the excluded phenotype. Moreover, an association between PD-L1 expression, immunophenotype, and ISUP grade was observed. Thus, our observations were consistent with those of previous reports [18,19,20,21,22,23].

In most other tumors, including melanoma, colorectal cancer, and lung cancer, the immunogenic features of the TIME (tumor infiltration by CD8+ T cells; inflamed phenotype) are considered to be markers of good prognosis [24,25], while they are associated with poor prognosis in RCC [20]. Although this is thought to be due to the strong association between the TIME and pathological grade in RCC, a more detailed study on the TIME is needed.

We examined the association between the TIME and synchronous/metachronous disease, including other pathological features. PD-L1 expression, WHO/ISUP grade, and LVI were associated with synchronous/metachronous disease. To our knowledge, this is the first report to show this association, and we consider the results to be a recommendation to evaluate each of the TIME and pathological features for better understanding of synchronous/metachronous RCC.

### 4.3. Clinical Importance of Time to Metastasis and TIME

We observed that PD-L1 positivity differed between metastases within 12 months and metastases > 12 months. However, this observation was not very obvious in the distribution of the immunophenotypes. 

A trend of differential prevalence within and after 12 months of time to metastasis was also observed for pathological features other than PD-L1 expression. LVI and WHO/ISUP grade 1/2, which were selected in the synchronous/metachronous logistic model, showed similar trends as PD-L1 expression. All SDs with a 12-month cutoff point for the time to metastasis supported this observation.

These results suggest that even if a metastasis is categorized as metachronous, if the time to metastasis is <12 months, the tumor environment is close to that of a synchronous metastasis. Specifically, the tumor environment of a metastasis that develops within 12 months is immunologically more tumor-infiltrating lymphocyte (TIL)-infiltrated and PD-L1-expressing and pathologically of a higher grade than the tumor environment of a metastasis that develops after 12 months.

The MSKCC risk classification in the cytokine era [3] and the IMDC risk classification in the targeted therapy era [4] are most widely used in the prognostic factor models of mRCC. Both classifications include “time from diagnosis to treatment of less than 12 months” as a risk factor. The “time from diagnosis to treatment” is “the time from initial diagnosis to confirmation of metastasis.” Therefore, the observation that both PD-L1 positivity and pathological features tended to differ at about 12 months in our study might support the setting of 12 months as the cutoff point in IMDC and MSKCC. The IMDC/MSKCC criteria do not include pathological grade or other pathological features that are considered as prognostic factors, but this may be because of employing the 12-month cutoff as a factor to explain pathological features.

### 4.4. Clinical Importance of TIME in the Current IO Era

The rapid evolution from TKI to immunotherapy with checkpoint inhibitors has dramatically changed treatment outcomes in mRCC [9]. The results from different studies also suggest a trend toward different subgroups of patients benefiting from checkpoint inhibitors and VEGFRi. In the post hoc analysis of the CheckMate 214 trial (nivolumab plus ipilimumab vs. sunitinib), although the overall response rate (ORR) of sunitinib decreased as the number of risk factors increased (50–16%), the ORR of nivolumab plus ipilimumab was consistent across any number of IMDC risk factors (ranging from 39% to 44%) [26]. Furthermore, while longer OS and a higher ORR were observed with nivolumab plus ipilimumab than with sunitinib across tumor PD-L1 expression levels, the magnitude of benefit was higher in the population with 1% or greater PD-L1 expression [7]. In addition, in RCC with sarcomatoid histology, which is known to be associated with a poor response to VEGFRi, both avelumab plus axitinib and nivolumab plus ipilimumab showed better efficacy outcomes compared with sunitinib in the post hoc analysis [8,27]. 

The post hoc biomarker analysis of iMmotion150 (atezolizumab plus bevacizumab, atezolizumab vs. sunitinib) and iMmotion151 (atezolizumab plus bevacizumab vs. sunitinib) based on gene expression associated with “tumor angiogenesis,” “pre-existing immunity,” and “immunosuppressive myeloid inflammation” showed that these signatures were differentially associated with progression-free survival (PFS)/OS across treatments [21,28]. Favorable trends of PFS/OS for atezolizumab plus bevacizumab were observed in tumors characterized by high T-effector gene expression, but not in tumors characterized by high angiogenic gene expression [21,28]. The immunophenotype evaluated in our study assessed the localization of TILs relative to the tumor area [20] and may represent the status of the pre-existing immunity. 

In summary, IMDC risk, pathological features (sarcomatoid histology and tumor angiogenesis), and the TIME (status of pre-existing immunity, PD-L1 expression, and T-effector gene expression) are considered key factors in selecting 1L treatment for mRCC. With the advent of IO combination therapies, it is important to select treatment according to tumor type [29], but the TIME has not yet been evaluated in clinical practice. 

Our study showed that the TIME of synchronous metastasis or tumors with shorter time to recurrence was immunologically more TIL-infiltrated and PD-L1-expressing and pathologically of a higher grade than that of metachronous metastases. We also observed that the IMDC risk distribution at the time of the 1L therapy was worse in SG than in MG (favorable/intermediate/poor: 3.6%/64.8%/31.6% vs. 39.5%/51%/9.2%), and the OS under VEGFRi therapy was shorter in SG than in MG (median OS: 29.5 months vs. 44.2 months, HR: 0.74 [95% CI: 0.60, 0.91]). Thus, our study showed that the timing of metastasis is related to the TIME of the primary tumor and OS under VEGFRi treatment. As immune markers have not been evaluated in the current clinical practice of RCC, it is possible that the timing of metastasis reflects the TIME and may be useful in selecting 1L treatment.

### 4.5. Limitations

This study had several limitations. First, the patients included in this study underwent nephrectomy and received VEGFRi as 1L systemic therapy. However, since we evaluated the time to metastasis before systemic therapy and not OS, the results were not affected by systemic therapy. The response to IO drugs is unknown in this cohort. In this study, surgical specimens were used in all patients, and about half of the specimens were from cytoreductive nephrectomy. However, considering the results of the CARMENA study [30] and the fact that there are a certain number of patients who cannot undergo surgery, we thought it necessary to examine the use of needle biopsy specimens. Second, we analyzed the immune status of the primary tumors. The TIME of metastatic lesions, which is the target of systemic therapy, especially in the MG, might be different from the TIME of the primary lesions at the time of nephrectomy. Only patients who underwent radical nephrectomy were included in the SG group. The TIME of SG patients who cannot undergo radical nephrectomy might be different from that of patients who can undergo cytoreductive nephrectomy. Third, owing to the retrospective nature of the study, unmeasured confounding factors and/or selection bias could have affected the study results. Fourth, PD-L1 expression status was analyzed in tumor-infiltrating immune cells but not in tumor cells. Factors that were not considered in this study, such as PD-L1 expression status in tumor cells and gene expression [31,32] or mutation analysis, should be investigated in the future.

## 5. Conclusions

In conclusion, this is the first study to examine the differences in the TIME characteristics of primary lesions in patients with mRCC based on the time to metastasis. We demonstrated that PD-L1 expression in tumor-infiltrating immune cells and immunophenotypes of the primary lesions differed according to the time to recurrence. A synchronous or shorter time to recurrence was associated with increased PD-L1 expression in tumor-infiltrating immune cells, more inflamed and excluded phenotypes, and fewer desert phenotypes. In the IO era, an understanding of the TIME of primary lesions might provide useful insights regarding the choice of 1L systemic therapy in patients with mRCC.

## Figures and Tables

**Figure 1 cancers-14-05258-f001:**
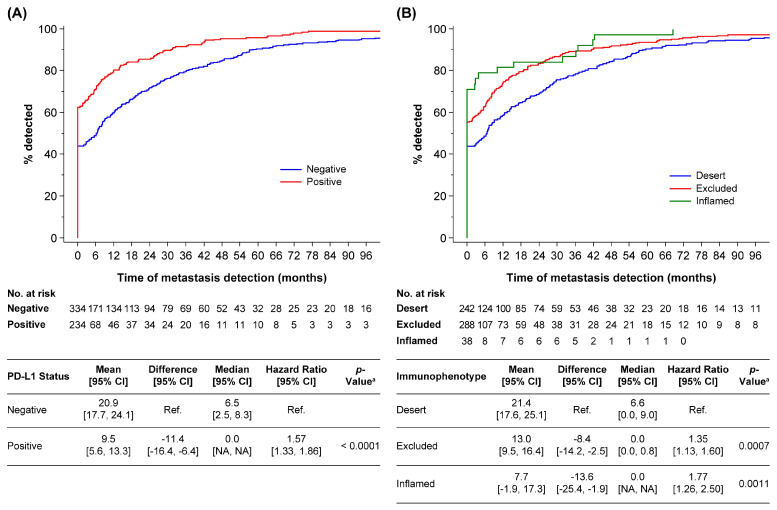
Kaplan-Meier curve of time to metastasis by (**A**) PD-L1 status or (**B**) immunophenotype. ^a^
*p*-values were calculated using the Wald test for a parameter of the univariate Cox proportional hazards model. CI, confidence interval; NA, not applicable; PD-L1, programmed death ligand 1; ref., reference.

**Figure 2 cancers-14-05258-f002:**
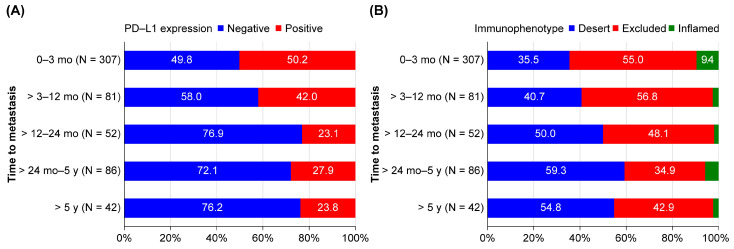
Distribution of immunological features by time to metastasis (100% stacked bar chart). (**A**) PD-L1 status by time from initial diagnosis to metastasis. (**B**) Immunophenotype by time from initial diagnosis to metastasis. PD-L1, programmed death ligand 1.

**Table 1 cancers-14-05258-t001:** Baseline characteristics at the time of initial diagnosis of synchronous and metachronous metastatic renal cell carcinoma.

Characteristic, n (%)	Synchronous ^a^ (N = 307)	Metachronous ^b^ (N = 261)	Total (N = 568)	*p*-Value ^c^	Standardized Difference
**Sex ^d^**					
Male	238 (77.5)	196 (75.1)	434 (76.4)	0.497	0.1
Female	69 (22.5)	65 (24.9)	134 (23.6)		−0.1
**Age**					
Mean (standard deviation)	63.2 (10.9)	62.8 (10.3)	63.0 (10.6)	0.623	0.0
Median [range]	64.0 [23, 87]	64.0 [30, 85]	64.0 [23, 87]		
**Age category ^d^**					
<65 y	164 (53.4)	140 (53.6)	304 (53.5)	0.473	0.0
≥65 and <75 y	94 (30.6)	88 (33.7)	182 (32.0)		−0.1
≥75 y	49 (16.0)	33 (12.6)	82 (14.4)		0.1
**Histology ^d^**					
Clear cell	286 (93.2)	241 (92.3)	527 (92.8)	0.706	0.0
Non–clear cell	21 (6.8)	20 (7.7)	41 (7.2)		0.0
**Sarcomatoid component ^d^**					
Absent	257 (83.7)	247 (94.6)	504 (88.7)	<0.0001	−0.4
Present	50 (16.3)	14 (5.4)	64 (11.3)		0.4
**Growth pattern ^d^**					
Expansive	97 (31.6)	110 (42.1)	207 (36.4)	0.021	−0.2
Infiltrative	87 (28.3)	55 (21.1)	142 (25.0)		0.2
Indeterminable	123 (40.1)	96 (36.8)	219 (38.6)		0.1
**Fuhrman grade ^d^**					
Grade 1/2	78 (25.4)	107 (41.0)	185 (32.6)	<0.0001	−0.3
Grade 3	154 (50.2)	126 (48.3)	280 (49.3)		0.0
Grade 4	75 (24.4)	28 (10.7)	103 (18.1)		0.4
**WHO/ISUP grade ^d^**					
Grade 1/2	91 (29.6)	124 (47.5)	215 (37.9)	<0.0001	−0.4
Grade 3	117 (38.1)	98 (37.5)	215 (37.9)		0.0
Grade 4	99 (32.2)	39 (14.9)	138 (24.3)		0.4
**Necrosis ^d^**					
Absent	150 (48.9)	169 (64.8)	319 (56.2)	0.0007	−0.3
Present	156 (50.8)	91 (34.9)	247 (43.5)		0.3
Indeterminable	1 (0.3)	1 (0.4)	2 (0.4)		0.0
**Lymphovascular invasion ^d^**					
Absent	195 (63.5)	194 (74.3)	389 (68.5)	0.0074	−0.2
Present	94 (30.6)	50 (19.2)	144 (25.4)		0.3
Indeterminable	18 (5.9)	17 (6.5)	35 (6.2)		0.0
**TIME**					
**PD-L1 expression**					
IC0	153 (49.8)	181 (69.3)	334 (58.8)	<0.0001	−0.4
IC1	88 (28.7)	56 (21.5)	144 (25.4)		0.2
IC2	38 (12.4)	16 (6.1)	54 (9.5)		0.2
IC3	28 (9.1)	8 (3.1)	36 (6.3)		0.3
**PD-L1 expression ^d^**					
Negative ^e^	153 (49.8)	181 (69.3)	334 (58.8)	<0.0001	−0.4
Positive ^f^	154 (50.2)	80 (30.7)	234 (41.2)		0.4
**Immunophenotype ^d^**					
Desert	109 (35.5)	133 (51.0)	242 (42.6)	0.0001	−0.3
Excluded	169 (55.0)	119 (45.6)	288 (50.7)		0.2
Inflamed	29 (9.4)	9 (3.4)	38 (6.7)		0.2

^a^ Defined as metastasis ≤ 3 months of initial diagnosis of renal cell carcinoma. ^b^ Defined as metastasis diagnosed > 3 months after initial diagnosis. ^c^
*p*-values were calculated using the chi-square test for categorical variables and Kruskal-Wallis test for continuous variables. ^d^ Candidate explanatory variables in the multivariable logistic regression model of metachronous/synchronous renal cell carcinoma; results are shown in Table 2. ^e^ Defined as IC0. ^f^ Defined as IC1/2/3. ISUP, International Society of Urologic Pathologists; PD-L1, programmed death ligand 1; TIME, tumor immune microenvironment; WHO, World Health Organization.

**Table 2 cancers-14-05258-t002:** Multivariable logistic regression analysis of synchronous/metachronous metastasis.

Selected Variable ^a^	Definition of OR ^b^	Adjusted OR ^b^ [95% CI]	*p*-Value ^c^
PD-L1 expression	Positive/Negative	1.76 [1.22, 2.55]	0.0026
WHO/ISUP grade	Grade 3/Grades 1, 2	1.38 [0.93, 2.05]	0.110
	Grade 4/Grades 1, 2	2.58 [1.59, 4.20]	0.0001
Lymphovascular invasion	Present/Absent	1.60 [1.06, 2.40]	0.024

^a^ All candidate explanatory variables are described in Table 1 (sex, age category at initial diagnosis, PD-L1 expression, immunophenotype, histology, sarcomatoid component, growth pattern, Fuhrman grade, WHO/ISUP grade, necrosis, and lymphovascular invasion). Explanatory variable addition and removal were based on stepwise selection with α = 0.05. ^b^ Odds for synchronous/metachronous. ^c^
*p*-value was calculated using the Wald test for a parameter of the multivariable logistic regression model. CI, confidence interval; ISUP, International Society of Urologic Pathologists; OR, odds ratio; PD-L1, programmed death ligand 1; WHO, World Health Organization.

**Table 3 cancers-14-05258-t003:** Baseline characteristics by time from initial diagnosis to metastasis.

Characteristic, n (%)	Time from Initial Diagnosis to Metastasis					Total (N = 568)	*p*-Value ^a^
	≤3 mo (N = 307)	>3–12 mo (N = 81)	>12–24 mo (N = 52)	>24 mo–5 y (N = 86)	>5 y (N = 42)		
**Sex**							
Male	238 (77.5)	60 (74.1)	39 (75.0)	63 (73.3)	34 (81.0)	434 (76.4)	0.832
Female	69 (22.5)	21 (25.9)	13 (25.0)	23 (26.7)	8 (19.0)	134 (23.6)	
**Age**							
Mean (standard deviation)	63.2 (10.9)	63.2 (11.4)	62.9 (10.0)	63.7 (9.8)	60.2 (9.5)	63.0 (10.6)	0.281
Median [range]	64.0 [23, 87]	65.0 [30, 85]	63.0 [32, 81]	66.0 [35, 81]	61.0 [38, 84]	64.0 [23, 87]	
**Age category**							
<65 y	164 (53.4)	39 (48.1)	31 (59.6)	41 (47.7)	29 (69.0)	304 (53.5)	0.179
≥65 and <75 y	94 (30.6)	27 (33.3)	15 (28.8)	34 (39.5)	12 (28.6)	182 (32.0)	
≥75 y	49 (16.0)	15 (18.5)	6 (11.5)	11 (12.8)	1 (2.4)	82 (14.4)	
**Histology**							
Clear cell	286 (93.2)	74 (91.4)	46 (88.5)	79 (91.9)	42 (100.0)	527 (92.8)	0.274
Non–clear cell	21 (6.8)	7 (8.6)	6 (11.5)	7 (8.1)	0 (0.0)	41 (7.2)	
**Sarcomatoid component**							
Absent	257 (83.7)	74 (91.4)	46 (88.5)	85 (98.8)	42 (100.0)	504 (88.7)	0.0002
Present	50 (16.3)	7 (8.6)	6 (11.5)	1 (1.2)	0 (0.0)	64 (11.3)	
**Growth pattern**							
Expansive	97 (31.6)	23 (28.4)	23 (44.2)	45 (52.3)	19 (45.2)	207 (36.4)	0.0044
Infiltrative	87 (28.3)	27 (33.3)	10 (19.2)	11 (12.8)	7 (16.7)	142 (25.0)	
Indeterminable	123 (40.1)	31 (38.3)	19 (36.5)	30 (34.9)	16 (38.1)	219 (38.6)	
**Fuhrman grade**							
Grade 1/2	78 (25.4)	21 (25.9)	20 (38.5)	42 (48.8)	24 (57.1)	185 (32.6)	<0.0001
Grade 3	154 (50.2)	46 (56.8)	24 (46.2)	40 (46.5)	16 (38.1)	280 (49.3)	
Grade 4	75 (24.4)	14 (17.3)	8 (15.4)	4 (4.7)	2 (4.8)	103 (18.1)	
**WHO/ISUP grade**							
Grade 1/2	91 (29.6)	21 (25.9)	26 (50.0)	51 (59.3)	26 (61.9)	215 (37.9)	<0.0001
Grade 3	117 (38.1)	43 (53.1)	13 (25.0)	28 (32.6)	14 (33.3)	215 (37.9)	
Grade 4	99 (32.2)	17 (21.0)	13 (25.0)	7 (8.1)	2 (4.8)	138 (24.3)	
**Necrosis**							
Absent	150 (48.9)	37 (45.7)	35 (67.3)	63 (73.3)	34 (81.0)	319 (56.2)	<0.0001
Present	156 (50.8)	43 (53.1)	17 (32.7)	23 (26.7)	8 (19.0)	247 (43.5)	
Indeterminable	1 (0.3)	1 (1.2)	0 (0.0)	0 (0.0)	0 (0.0)	2 (0.4)	
**Lymphovascular invasion**							
Absent	195 (63.5)	54 (66.7)	43 (82.7)	68 (79.1)	29 (69.0)	389 (68.5)	0.0081
Present	94 (30.6)	23 (28.4)	7 (13.5)	13 (15.1)	7 (16.7)	144 (25.4)	
Indeterminable	18 (5.9)	4 (4.9)	2 (3.8)	5 (5.8)	6 (14.3)	35 (6.2)	
**TIME**							
**PD-L1 expression**							
IC0	153 (49.8)	47 (58.0)	40 (76.9)	62 (72.1)	32 (76.2)	334 (58.8)	0.0005
IC1	88 (28.7)	22 (27.2)	9 (17.3)	18 (20.9)	7 (16.7)	144 (25.4)	
IC2	38 (12.4)	9 (11.1)	3 (5.8)	3 (3.5)	1 (2.4)	54 (9.5)	
IC3	28 (9.1)	3 (3.7)	0 (0.0)	3 (3.5)	2 (4.8)	36 (6.3)	
**PD-L1 expression**							
Negative ^b^	153 (49.8)	47 (58.0)	40 (76.9)	62 (72.1)	32 (76.2)	334 (58.8)	<0.0001
Positive ^c^	154 (50.2)	34 (42.0)	12 (23.1)	24 (27.9)	10 (23.8)	234 (41.2)	
**Immunophenotype**							
Desert	109 (35.5)	33 (40.7)	26 (50.0)	51 (59.3)	23 (54.8)	242 (42.6)	0.0008
Excluded	169 (55.0)	46 (56.8)	25 (48.1)	30 (34.9)	18 (42.9)	288 (50.7)	
Inflamed	29 (9.4)	2 (2.5)	1 (1.9)	5 (5.8)	1 (2.4)	38 (6.7)	

^a^*p*-values were calculated using the chi-square test for categorical variables and Kruskal-Wallis test for continuous variables. ^b^ Defined as IC0. ^c^ Defined as IC1/2/3. ISUP, International Society of Urologic Pathologists; PD-L1, programmed death ligand 1; TIME, tumor immune microenvironment; WHO, World Health Organization.

**Table 4 cancers-14-05258-t004:** Clinical characteristics at the time of first-line treatment of synchronous and metachronous metastatic renal cell carcinoma.

Characteristic, n (%)	Synchronous ^a^ (N = 307)	Metachronous ^b^ (N = 261)	Total (N = 568)	*p*-Value ^c^	Standardized Difference
**Sex**					
Male	238 (77.5)	196 (75.1)	434 (76.4)	0.497	0.1
Female	69 (22.5)	65 (24.9)	134 (23.6)		−0.1
**Age (year)**					
Mean (standard deviation)	63.8 (10.9)	66.4 (10.3)	65.0 (10.7)	0.0032	−0.2
Median [range]	64.0 [23, 87]	68.0 [31, 89]	66.0 [23, 89]		
**Age category**					
<65 y	155 (50.5)	102 (39.1)	257 (45.2)	0.021	0.2
≥65 and <75 y	100 (32.6)	100 (38.3)	200 (35.2)		−0.1
≥75 y	52 (16.9)	59 (22.6)	111 (19.5)		−0.1
**ECOG PS category**					
0, 1	251 (81.8)	212 (81.2)	463 (81.5)	0.400	0.0
≥2	38 (12.4)	27 (10.3)	65 (11.4)		0.1
Unknown	18 (5.9)	22 (8.4)	40 (7.0)		−0.1
**IMDC risk group**					
Favorable	11 (3.6)	103 (39.5)	114 (20.1)	< 0.0001	−1.0
Intermediate	199 (64.8)	134 (51.3)	333 (58.6)		0.3
Poor	97 (31.6)	24 (9.2)	121 (21.3)		0.6
**White blood cells**					
≤ULN	253 (82.4)	231 (88.5)	484 (85.2)	0.107	−0.2
>ULN	49 (16.0)	26 (10.0)	75 (13.2)		0.2
Indeterminable	5 (1.6)	4 (1.5)	9 (1.6)		0.0
**Neutrophils**					
≤ULN	219 (71.3)	187 (71.6)	406 (71.5)	0.144	0.0
>ULN	63 (20.5)	42 (16.1)	105 (18.5)		0.1
Indeterminable	25 (8.1)	32 (12.3)	57 (10.0)		−0.1
**Neutrophil–lymphocyte ratio**					
<2.9	135 (44.0)	131 (50.2)	266 (46.8)	0.021	−0.1
≥2.9	147 (47.9)	97 (37.2)	244 (43.0)		0.2
Indeterminable	25 (8.1)	33 (12.6)	58 (10.2)		−0.1
**CRP (mg/dL)**					
<0.3	92 (30.0)	129 (49.4)	221 (38.9)	<0.0001	−0.4
≥0.3	199 (64.8)	114 (43.7)	313 (55.1)		0.4
Indeterminable	16 (5.2)	18 (6.9)	34 (6.0)		−0.1
**Hemoglobin**					
≥LLN	112 (36.5)	153 (58.6)	265 (46.7)	<0.0001	−0.5
<LLN	190 (61.9)	104 (39.8)	294 (51.8)		0.5
Indeterminable	5 (1.6)	4 (1.5)	9 (1.6)		0.0
**Platelets**					
≤ULN	258 (84.0)	243 (93.1)	501 (88.2)	0.0020	−0.3
>ULN	44 (14.3)	14 (5.4)	58 (10.2)		0.3
Indeterminable	5 (1.6)	4 (1.5)	9 (1.6)		0.0
**Corrected serum calcium (mg/dL)**					
≤10	250 (81.4)	223 (85.4)	473 (83.3)	0.058	−0.1
>10	39 (12.7)	18 (6.9)	57 (10.0)		0.2
Indeterminable	18 (5.9)	20 (7.7)	38 (6.7)		−0.1
**Lactate dehydrogenase**					
≤ULN × 1.5	273 (88.9)	241 (92.3)	514 (90.5)	0.370	−0.1
>ULN × 1.5	19 (6.2)	12 (4.6)	31 (5.5)		0.1
Indeterminable	15 (4.9)	8 (3.1)	23 (4.0)		0.1
**Albumin**					
≥LLN	95 (30.9)	128 (49.0)	223 (39.3)	<0.0001	−0.4
<LLN	199 (64.8)	123 (47.1)	322 (56.7)		0.4
Indeterminable	13 (4.2)	10 (3.8)	23 (4.0)		0.0

^a^ Defined as metastasis ≤ 3 months of initial diagnosis of renal cell carcinoma. ^b^ Defined as metastasis diagnosed > 3 months after initial diagnosis of renal cell carcinoma. ^c^
*p*-values were calculated using the chi-square test for categorical variables and Kruskal-Wallis test for continuous variables. CRP, C-reactive protein; ECOG, Eastern Cooperative Oncology Group; IMDC, International Metastatic Renal Cell Carcinoma Database Consortium; LLN, lower limit of normal; PS, performance status; ULN, upper limit of normal.

## Data Availability

Qualified researchers may request access to individual patient-level data through the clinical study data request platform (www.clinicalstudydatarequest.com). For further details on Chugai’s Data Sharing Policy and how to request access to related clinical study documents, see here (www.chugai-pharm.co.jp/english/profile/rd/ctds_request.html).

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
