# Peer review of "The Association of Tumor Immune Microenvironment of the Primary Lesion with Time to Metastasis in Patients with Renal Cell Carcinoma: A Retrospective Analysis"

_cancers, 2022, doi:10.3390/cancers14215258_

Round 1
Reviewer 1 Report
The authors investigated the classification of the immune environment in patients who underwent nephrectomy in a cohort of patients with metastatic renal cancer treated with tyrosine kinase inhibitor as first-line therapy. They demonstrated that the immune environment, which they call tumor immune microenvironment (TIME), differs between synchronous and metachronous metastasis groups.
I think it is very excellent and well described. Please comment on the following points.
1. After the reports such as the CARMENA trial (Méjean A, et al. N Engl J Med. 2018;379:417-427), the significance of cytoreductive nephrectomy in metastatic cases is getting decreased. In the future, we will need the pathological evaluation of tumors in needle biopsies. Could the findings of this study be applied to needle biopsy specimens?
2. The findings shown in Figure 2 suggest that there seems to be a difference at 12 months. Although the authors divide metachronous by 3 months, I think it would be simpler to divide by 12 months, as shown in Table S2. As the authors mentioned in the main body, I think it would be better and simpler to divide them into synchronous and metachronous at 12 months.
3. There is mention of two central pathologists. One is listed as an author but the other is not. I think it would be better to include it in the authorship or at least in the acknowledgment.
Author Response
- After the reports such as the CARMENA trial (Méjean A, et al. N Engl J Med. 2018;379:417-427), the significance of cytoreductive nephrectomy in metastatic cases is getting decreased. In the future, we will need the pathological evaluation of tumors in needle biopsies. Could the findings of this study be applied to needle biopsy specimens?
- Response1:As we described in "2.Materials and Methods (Only patients whose formalin-fixed paraffin-embedded nephrectomy specimens could be obtained were registered in this retrospective study.)", this study enrolled only patients with surgical specimens. Therefere we can't discuss biopsy samples in this paper.
Based on your comment, I have added that point to the "Limitation". Please find the attached revised manuscript. (the revised manuscript L406-410)
- Response1:As we described in "2.Materials and Methods (Only patients whose formalin-fixed paraffin-embedded nephrectomy specimens could be obtained were registered in this retrospective study.)", this study enrolled only patients with surgical specimens. Therefere we can't discuss biopsy samples in this paper.
- The findings shown in Figure 2 suggest that there seems to be a difference at 12 months. Although the authors divide metachronous by 3 months, I think it would be simpler to divide by 12 months, as shown in Table S2.
As the authors mentioned in the main body, I think it would be better and simpler to divide them into synchronous and metachronous at 12 months.- Response2:We chose the 3-month cut-off of the time to metastasis for the definition of synchronous/metachronous based on the previous report, therefore we would like to keep the current construct. In addition,this study didn't aim to examine the validity of 3months as a cutoff value.
As you indicated, we found 12 months to be an important cutoff value when we consider characteristics of TIME, therefore further investigation is warrant.
- Response2:We chose the 3-month cut-off of the time to metastasis for the definition of synchronous/metachronous based on the previous report, therefore we would like to keep the current construct. In addition,this study didn't aim to examine the validity of 3months as a cutoff value.
- There is mention of two central pathologists. One is listed as an author but the other is not. I think it would be better to include it in the authorship or at least in the acknowledgment.
- Response3:Thank you very much for your comment, we added the pathologist in the Acknowledge (the revised manuscript L456-458).

Reviewer 2 Report
I consider the present manuscript having an interesting topic, with a clear and easy to be individualized on literature research title. The abstract is well structured.
The Introduction is too short, with few data about RCC physiopathology. I recommend a larger description of TIME and the therapy for ccRCC in this section.
Material and Methods are well conceived, to make the study reproducible. The Results cover all the required fields.
Discussions are well conceived and support the Results, but some more references should be added: eg. line 343, line 353
The Conclusions reflect the idea of the title and the manuscript presents a recent bibliography, but with a small number of titles.
Author Response
- The Introduction is too short, with few data about RCC physiopathology. I recommend a larger description of TIME and the therapy for ccRCC in this section.
- Response1: We updated current RCC treatment and added description of TIME and features of TIME in RCC.
We added description in the introductioon, and 4 references (the revised manuscript L79-82, L84-89).
- Response1: We updated current RCC treatment and added description of TIME and features of TIME in RCC.
- Discussions are well conceived and support the Results, but some more references should be added: eg. line 343, line 353
- Response2: Regarding line343/353 (12 months cutoff), we added two references (31,32) in the limitation part (the revised manuscript L420).
- The Conclusions reflect the idea of the title and the manuscript presents a recent bibliography, but with a small number of titles.
- Response3: We have added six references regarding above response.

Reviewer 3 Report
The work entitled The Association of Tumor Immune Microenvironment of the Primary Lesion with Time to Metastasis in Patients with Renal Cell Carcinoma: A Retrospective Analysis is an interesting scientific study. My doubts are raised by the adopted interval of 3 months instead of the usual 6. Please add more detailed data about the first, and second-line therapy including immunotherapy and survival analysis.
Author Response
- My doubts are raised by the adopted interval of 3 months instead of the usual 6.
-
Response1: As we described in "2. Materials and Methods", we chose a 3months cutoff point based on the previous report.
There is no consensus on when the onset of metastasis should be considered meta-chronous—previous reports have referred to onset after the time of diagnosis (> 0 months) [13], after 3 months [2], and after 6 months [5]; the 3-month cutoff was chosen for this study.
-
- Please add more detailed data about the first, and second-line therapy including immunotherapy and survival analysis.
-
Response2: Since this paper focused on the association between time to metastasis and TIME of the primary tumor, I think your point is very important, but I did not provide details in this paper.
As we mentioned "2.2 Patients", this study is an exploratory study of using ARCHERY dataset. in the ARCHERY study, patients who recieved immunotherapy as first line were excluded. and detailed information of treatment was described in the manuscript[11].
-
